# Sequentially bidirectional gastrovascular flows in intricately branched digestive tract of planocerid flatworms

**Po-Chun Hsu, Yu-Hsun Chang, Yu-Ning Chiu, Wei-Ban Jie****\***

National Experimental High School at Hsinchu Science Park, Hsinchu City, Taiwan

\* wbjie@nehs.hc.edu.tw

## Abstract

Polyclad flatworms possess an intricately branched digestive system combining features of a gastrovascular cavity as well as a gastrointestinal tract. Nonetheless, the functions of this system remain unconfirmed, due to a lack of effective observation methods. This paper presents a novel staining method to facilitate the analysis of this highly branched digestive system. Video recordings obtained during ingestion revealed sequentially bidirectional gastrovascular flows and a corresponding occurrence of regular contractions. Tissue sections revealed that the contractions can be attributed to a radial arrangement of muscles around the gastrointestinal tract. The highly branched digestive system of the flatworm revealed evidence of bidirectional flow and sequential peristalsis, which may allow for a diet of greater diversity than is possible in animals with only a gastrovascular cavity. The proposed staining technique opens up new avenues for research on the digestive behavior of lower organisms.

## Introduction

The gastrovascular cavity, part of the basic body plan of cnidarians (e.g., jellyfish, corals, and hydras), is integral to digestion and nutrient distribution [1,2]. Shimizu et al. (2004) provided evidence of digestive movement in hydras, based on the way that a diffuse nerve net controls digestion and circulation through segmentation movement [3,4]. Although the gastric cavity of hydra is a blind sac, it is likely that segmentation movement in the cavity is a back-and-forth transference following contact with the end of the cavity [3].

Harmata et al. (2013) delved into gastrovascular flows in the cnidaria phylum, with a focus on hydroids and octocorals [5]. They described ciliary motion which was visible in image sequences, and sequentially bidirectional flow or simultaneously bidirectional flow separated by baffles [5]. In the absence of a true circulatory system, the gastrovascular cavity distributes partially digested material throughout the body. In hydroid species, myoepithelial contractions from the center of the colony drive sequentially bidirectional flows [5]. Using resin endocasts and 3D X-ray computed microtomography, Avian et al. (2022) characterized the branched gastrovascular system of *Rhizostoma pulmo* (Rhizostomatidae) as simultaneous inward and

**Data Availability Statement:** All relevant data are within the manuscript.

**Funding:** The author(s) received no specific funding for this work.

**Competing interests:** The authors have declared that no competing interests exist.

outward flows in a peculiar double hemi-canal structure. These findings suggest that in cnidarians, the various openings of the gastrovascular system may function as a through-gut apparatus [6].

Polyclad flatworms are free-living animals with branched intestines radiating into a peripheral network [7]. Previous research on the digestive structures of flatworms has focused on the gastrovascular cavity or the gastrointestinal tract. Similar to cnidarians, these flatworms feature an incomplete gut, described as a gastrovascular cavity [8]. In 1970, Koopowitz described the feeding behavior of the polyclad *Planocera gilchristi* collected from intertidal zones and held in a container with periwinkle (*Oxystele*) as prey. These organisms displayed complex feeding behaviors, such as prey recognition, stalking, capture, extraction, and swallowing [9]. Jennings (1957) described the digestive system of *Leptoplana tremellaris*, an acotylean flatworm that feeds on small animals, such as polychaetes, isopods, and amphipods. In that system, food entering the branched gut is partially broken down by enzymatic activities in the lumen before entering the cells [10]. In their work on *Pericelis flavomarginata*, a cotylean flatworm, Tsuyuki et al. (2020) reported that the intestinal branches gradually changed to a reddish hue at 1–2 hours after consuming red scaleworms, making the gut branches of the polyclad worm more visible [11]. Newman and Cannon (2003) described the predation behavior of *Planocera* sp., which can feast on cowrie in the waters off Micronesia [12]. Ritson-Williams et al. (2006) reported on the use of the deadly neurotoxin tetrodotoxin by a previously undescribed *Planocera* sp. for the capture of prey [13]. Taken together, these findings provide strong evidence that some polyclad flatworms are active predators.

In a previous analysis of *Planocerid* flatworms, our research team observed rhythmic contractions of the highly branched digestive system, associated with unknown ingestion patterns in the gastrovascular cavity. Newman and Cannon (2003) posited that the branches of the digestive system are lined with small muscular valves (sphincters) regulating flow in the form of contractions (peristaltic movement) to fill or clear the gut [12]. This raises questions pertaining to the peristaltic activity of polyclad flatworms compared to cnidarians as well as to the functional role of the small muscular valves along the branched digestive tracts. Nonetheless, there have been no further reports on this topic since 2003.

In the current study, we collected *Paraplanocera oligoglena* (Schmarda, 1859 [14]) from intertidal pools in Taiwan. *P. oligoglena* is a globally distributed polyclad flatworm [12,15,16], initially documented in Taiwan by Kato (1943) [17]. It has been featured in various field guide publications written in Chinese, [18]. Our objective was to unravel the mechanisms underlying the gastrovascular flows in the highly branched digestive system by employing the method proposed by Wells and Sebens (2017) for the staining of clams [19].

## Materials and methods

### Ethics statement

All the specimens used in this thesis do not involve endangered or invasive species and under the permission of the Kenting National Park or the other areas where specific permissions are not needed. These thesis had received the approval of National Taiwan Science Education Center (NTSEC) and the International Science and Engineering Fair (ISEF) 2024.

### Collection of specimens

*Paraplanocera oligoglena*, totally 13 samples were gathered from tidal pools in various locations of Taiwan, including 4 specimens from Longdong Bay (25˚06'48.1"N 121˚55'13.1"E), 6 specimens from Magang (25˚01'03.0"N 122˚00'00.8"E), and 3 specimens from Houbihu (21˚ 56'19.3"N 120˚44'44.1"E). The animals were held in a glass tank equipped with a circulating

filtration system with seawater salinity maintained at 32–34‰ at a temperature of 24–26˚C. The quality of the artificial seawater was maintained by regularly removing food residue and adding nitrifying bacteria.

## Bait preparation

This study prepared clams as bait in accordance with the methods outlined by Teng et al. (2022) for the study of oyster leech behavior [20]. Note that clams are not a natural prey of *Paraplanocera oligoglena* in the wild; however, we determined that market-sourced Asian hard clams (*Meretrix taiwanica*) provided a suitable food substitute for the observation of wild flatworms under laboratory conditions.

Clam meat removed via dissection was stained overnight using methylene blue and fluorescein in accordance with the staining technique proposed by Wells and Sebens (2017) to facilitate the *in vivo* observation of the digestive tract [19]. For extended video recording, the illuminated transparent glass tank and video camera JVC GZ-RX500BTW, Olympus TG6 were used. Video clips showing the ventral sides of the flatworms were captured after staining to observe the movement of food within the highly branched digestive system. All clips were recorded and transformed into the form of MP4 files as partially shown in S1 Video, and timelapse video was converted in 2 times play speed as shown in S2 Video.

## Image processing and tracking

Static images and video clips were subjected to image processing, including image transformation [21] and adaptive thresholding based on the Open Source Computer Vision Library (OpenCV) [22].

The flatworms were photographed during the post-feeding ingestion period on a petri dish lined with white paper beneath a uniform light source. The resulting images underwent background removal, conversion to grayscale, and noise filtering via Gaussian blurring. Adaptive thresholding was used to create a binary image in which blue-stained regions were depicted as white lines on a black background. The binary image was then converted into a BGR color image matching the dimensions of the original image. We also created a blank red image with identical dimensions, which was combined with the binary image to create a final image with the blue-stained regions highlighted as red lines.

We also captured video clips of the flatworms as stained food passed through the digestive tract. This involved moving the flatworms to the side of a transparent glass tank in front of a backlit white plastic divider providing a uniform light source. Selecting one specimen with the most pronounced process of food transport was used for the study. The recorded videos were sped up 10-fold for analytic convenience. We selected a region of interest (ROI) containing a small stretch of the digestive tract for continuous tracking. The lower and upper bounds of the object of interest, in this case, the stretch of tract in the selected ROI, were determined with Otsu's method [23]. Our use of CSRT (Discriminative Correlation Filter with Channel and Spatial Reliability) enabled the tracking of a selected area even under sudden movements. The stained-blue color in the frame was isolated via conversion to the HSV color space. The video clips then underwent graphing using Matplotlib [24] and Pandas [25] to determine the percentage of the stained areas in each frame as an indication of tract volume, wherein a higher percentage indicated tract expansion, while a lower percentage indicated contraction.

## Histological studies

Cross-sections of the tracts were obtained to observe the anatomy of the digestive tissue at the microscopic scale. Histologic analysis was performed in accordance with the standard

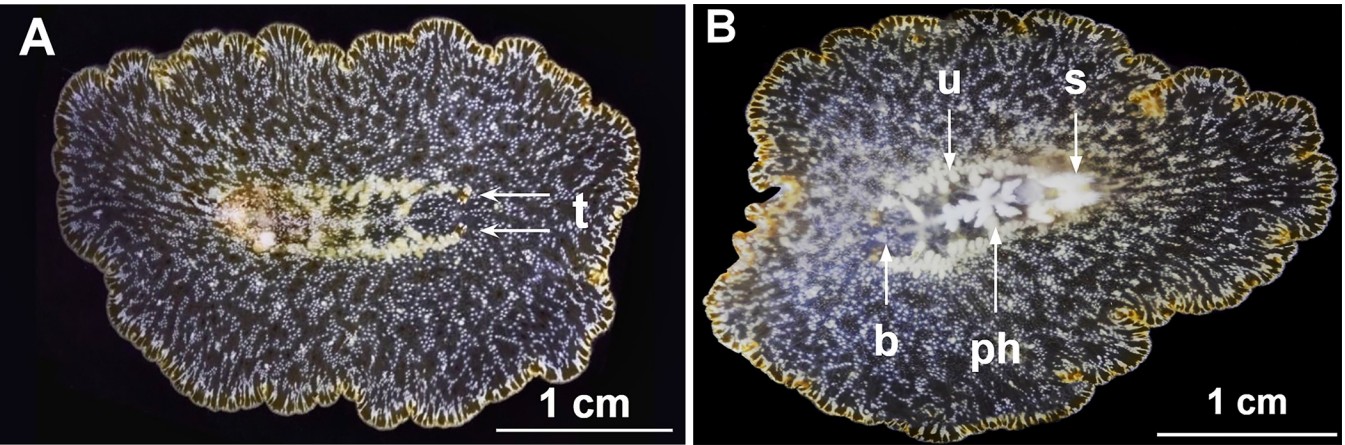

**Fig 1. *Paraplanocera oligoglena* (Schmarda, 1859) specimen collected at Longdong Bay, Taiwan on August 5th, 2022.** (A) Dorsal view with anterior end on the right; (B) Ventral view with anterior end on the left. b, brain; ph, pharynx; s, sexual organ; t, tentacle; u, uterus.

protocols outlined by Jie et al. (2013) [26]. Briefly, the tissues were fixed via embedding in paraffin following dehydration via serial passage using 30%, 50%, 70%, 95%, and 99.5% alcohol over a period of 2–4 hours.

## Results

*Paraplanocera oligoglena* (collected at Longdong Bay on Aug. 5th, 2022) presented an oval to circular body shape, which appeared somewhat translucent with a reticulated pattern of white and brown irregular spots over a light brown background color (Fig 1) [27]. The translucency and thin cross-section of the animals made it easy to observe the pharynx and main intestine from the ventral side. Note that this nearly transparent branched digestive system was so thin that it was barely detectable by the naked eye.

### Digestive system shown by stained food

The translucency of the body was also conducive to tracking the passage of digested food within the gastrointestinal tract (Fig 2A). Initial imaging results revealed that fluorescein staining had no effect, due to the autofluorescence of the flatworm, which eliminated any contrast

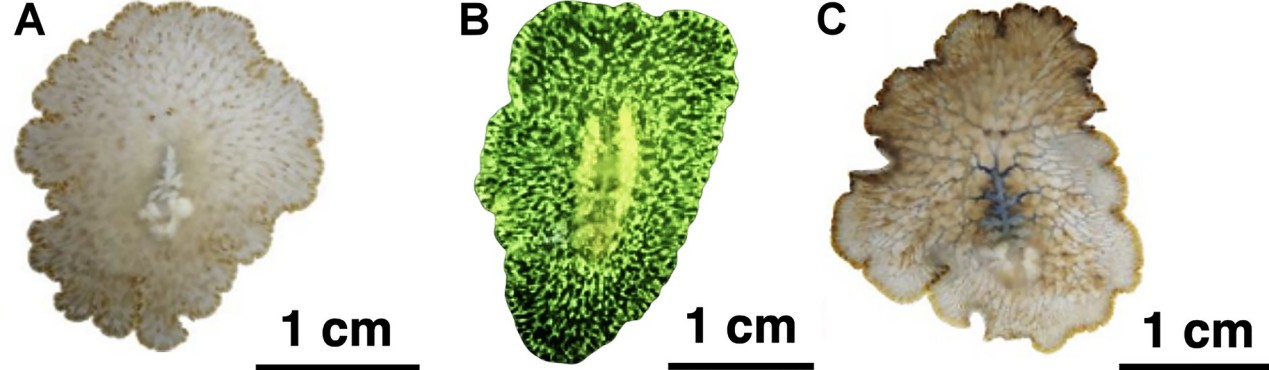

**Fig 2. Images of *Paraplanocera oligoglena*.** (A) Before staining, (B) after staining with fluorescein, and (C) after staining with methylene blue, which shows the digestive system comprising blind sacs, simple circulative loops, and extensive branching.

between the digestive system and rest of the body (Fig 2B). Thus, methylene blue was selected for all subsequent experiments (Fig 2C).

Fig 2C illustrates the digestive system of *Paraplanocera oligoglena* comprising blind sacs, simple circulative loops, and extensive branching. The blind sacs distributed along the digestive tracts resembled tubes that come to a dead end. The simple circulative loops connected adjacent branches to form ring-like formations distributed sporadically along the digestive tract (S1 and S2 Videos).

A defining characteristic of the digestive system of flatworms is the main intestine, which radiates as a series of branches from the pharynx toward the distal margin, resulting in fork-like bifurcations evenly distributed around most of the body. Note that the diameter of the digestive tract gradually diminishes as the branches bifurcate (Fig 3). Unlike the through-gut apparatus described by Avian et al. (2022) [6], the branched tract in this study appeared not to have any openings at the outermost ends.

The widest tract, originating from the central (i.e., pharyngeal) region, branched into a narrower secondary tract, which radiated into an even narrower tertiary tract. We devised a hierarchy describing the order of branches, wherein the branch originating from the pharynx was deemed the first-order ($n$), with the number increasing by 1 each time a bifurcation was passed (i.e., $n+1$, $n+2$...). Based on this system, it was determined that the narrowest tract was sixth-order branches. Note that the smallest branches were difficult to detect, such that the hierarchy may have extended even further (Fig 3).

As shown in Fig 4, the still images, Fig 2C were converted to grayscale and thresholded to identify areas stained with methylene blue, which were then highlighted using a red mask. Fig 4 illustrates a whole-body image with ROIs clearly illustrating the extreme complexity of the gastrovascular cavity.

## Post-staining active tracking: Gastrovascular flow

Shimizu et al. (2004) reported that the digestive system of hydra presents an esophageal-like segmented movement, similar to that of defecation [3]. In the current study, *Paraplanocera oligoglena* presented distinct digestive movements involving spontaneous contractions of the pharynx, gradually pushing the food along the digestive tract to the most remote margins (Fig 4).

Note that both static images and video recordings revealed sequentially bidirectional gastrovascular flows via peristalsis, which has not previously been reported. Unlike the segmented movement of hydra, the flatworms exhibited regular centrifugal sphincter contractions inward toward the outermost branches (i.e., highest order end) (Fig 5A and 5B). The organisms then exhibited centripetal movement from the outermost (narrowest) branches back to the pharynx (i.e., first-order digestive tract) (Fig 5C and 5D). This process exhibited a rhythmic pattern (S1 and S2 Videos). We obtained no evidence of flow originating from any of the blind sacs distributed along any of the tracts.

The tracker package in OpenCV was used to track the movement of food bidirectionally along the consecutive branches of orders n, n+1, n+2, and n+3. Matplotlib was then used to plot the stained area as a percentage of the selected region over time (S1 Table). The video clips revealed that the contractions began when the digestive tract of the flatworms narrowed. From this, we deduced that the wave trough of the curve represents the duration of the contraction, whereas the period between two peaks and wave troughs indicates the length of a contraction cycle (Fig 6).

As shown in Fig 7 (S2 Table), the ANOVA of the designated inward flows show no significant differences in consecutive n, n+1, n+2 branches in ANOVA (n = 30, d.f. = 2, F = 0.70,

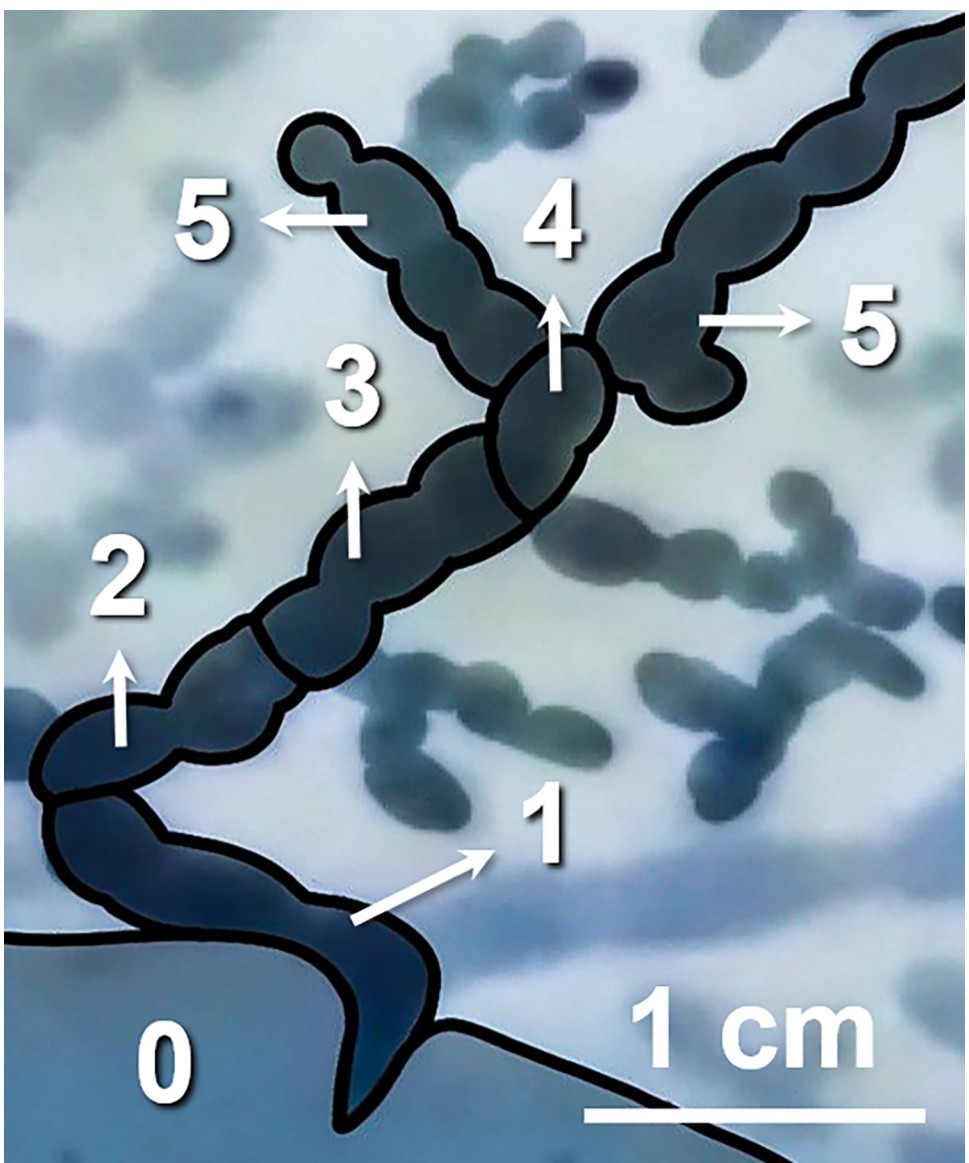

**Fig 3. Consecutive ordering of tract branches from the pharynx toward the distal margin.** The pharynx is indicated as 0, and then the numbers from 1 to 5 correspond to every segment of the digestive tract in each order.

p = 0.50). The designated outward flows also show no significant differences in consecutive n +2, n+1, n branches in ANOVA (n = 30, d.f. = 2, F = 0.39, p = 0.67). The frequency and speed of the contractions remained constant, regardless of the branch level (n+1, n+2. . .), which indicates that *P. oligoglena* does not rely exclusively on pharynx contraction for the transport of food. The segmented movements also indicate that the tract itself is involved in the contraction. We observed contractions occurring sequentially from one fixed position to another fixed position, resulting in peristalsis (rhythmic wave-like contractions), first in one direction and then in the opposite direction. Regardless of the direction, the movement remained consistent and sequentially bidirectional, albeit at an uneven rate. This is the first study to describe peristaltic segmented contractions and sequentially bidirectional movement in the gastrovascular cavity of polyclad flatworms.

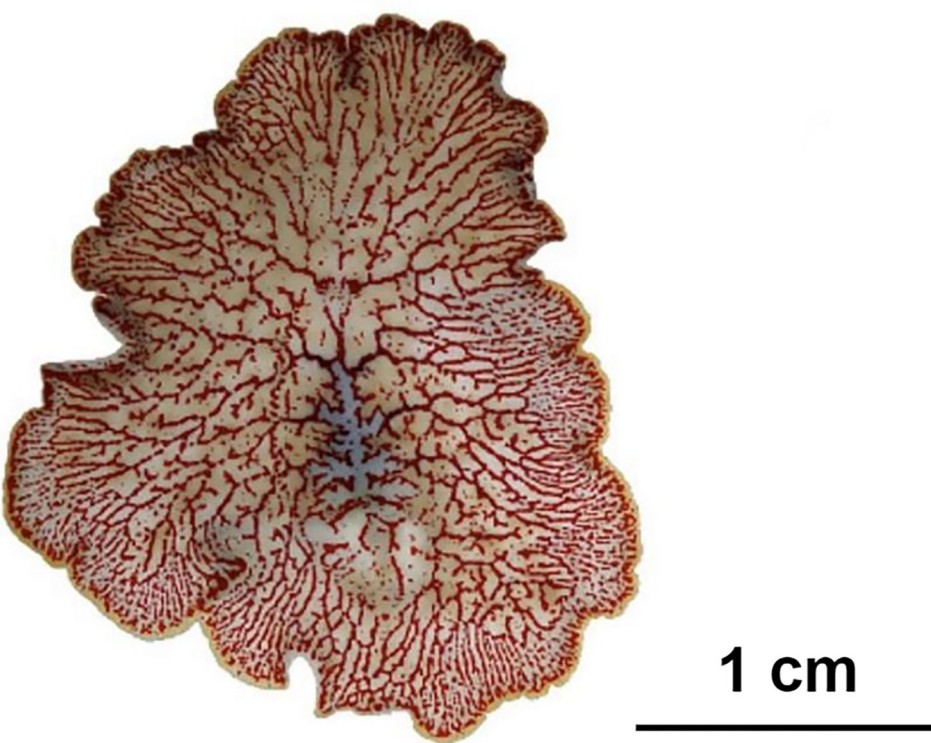

**Fig 4. Post-staining image of *Paraplanocera oligoglena* after image processing.** The red color indicates regions containing stained food. It shows a whole-body image with ROIs clearly illustrating the extreme complexity of the gastrovascular cavity.

As shown in Fig 8, the activity of the digestive system decreased dramatically after roughly one day and came to a complete stop within two days (S3 Table), after which the remaining food residue was moved back to the pharynx and expelled, in a manner similar to the termination of regular oscillatory behavior of post-feeding in isolated polyps [2] and the defecation reflex observed in hydra [3].

## Radial muscles inside the digestive tract

We also performed microscopic analysis of sections of the digestive tract to elucidate the constituent anatomical structures. We identified radial muscle structures evenly distributed along

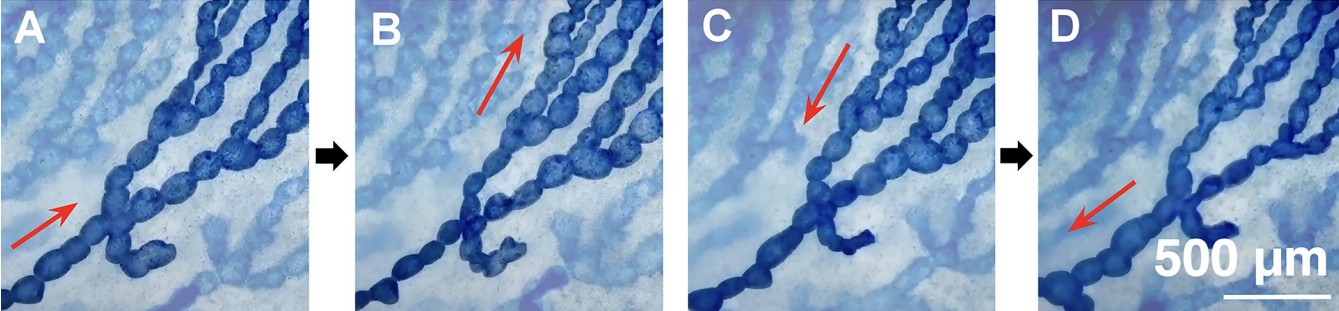

**Fig 5. Sequential video captures of bidirectional flow.** From (A) to (B) shows the inward gastrovascular flow and from (C) to (D) shows the outward flow in the same track. Red arrows indicate the direction of flow.

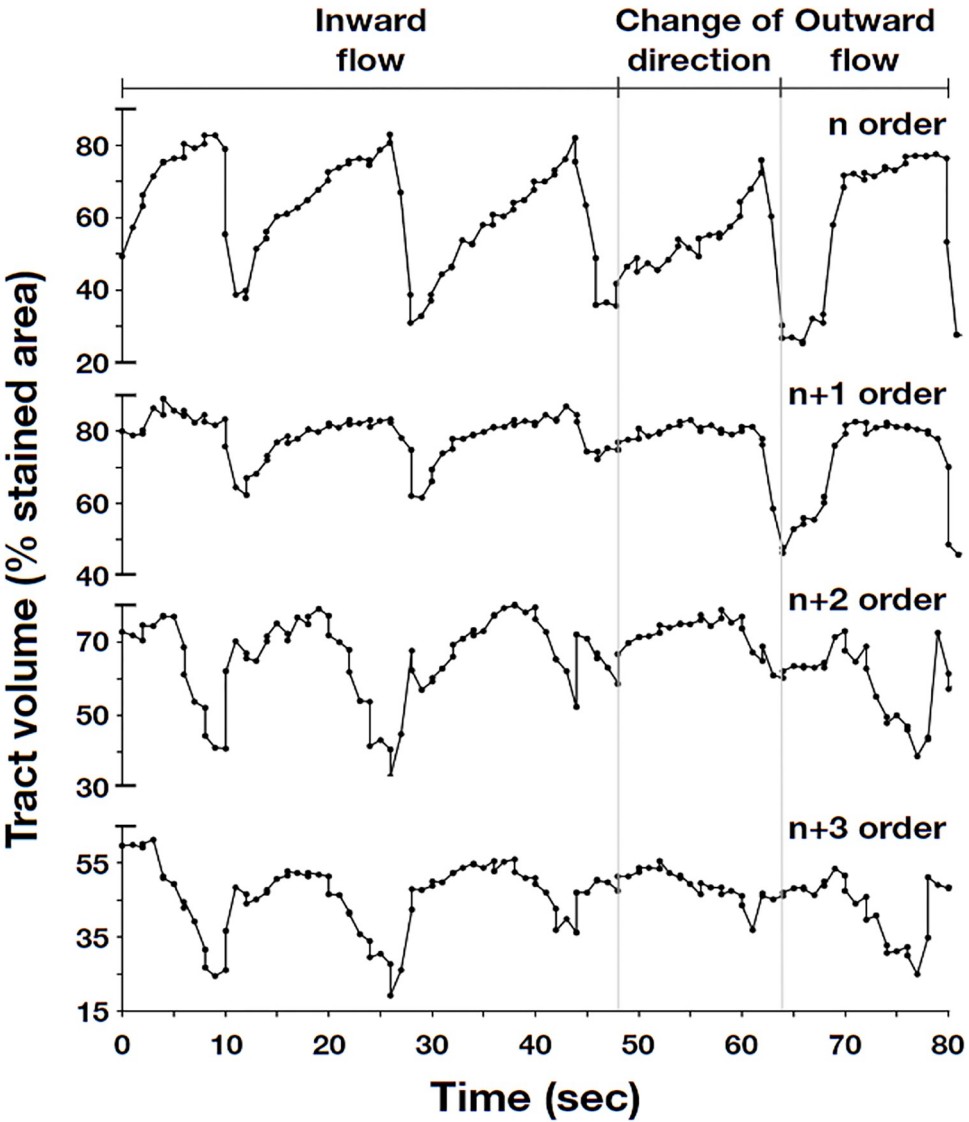

**Fig 6. Post-staining video tracking of dyed food passing through consecutive branches n, n+1, n+2, and n+3.** The timing of the events was as follows: Inward peristalsis (0 to 40 sec); change in the direction of gastrovascular flow (40 to 55 sec.); outward peristalsis (55 to 85 sec.). The wave trough of the curve indicates the duration of each contraction, while the period between two peaks and wave troughs represents a single peristalsis cycle.

the tract (Fig 9). This finding aligns with the description of small muscular valves along the branches of *Planocera* sp. proposed by Newman and Cannon in 2003 [12].

Furthermore, it was determined that the average distance between radial muscles was inversely proportional to the position of the sample within the hierarchical tract structure. In other words, the arrangement of radial muscles along narrower tracts (close to the distal margin) was denser than along larger tracts (close to the pharynx) (Fig 9B). From this, it can be inferred that the contractions of the radial muscles provide stable inward forces right from the utmost marginal branched digestive tract. Then, the sequentially bidirectional flows proceed back and forth within the branched digestive tract.

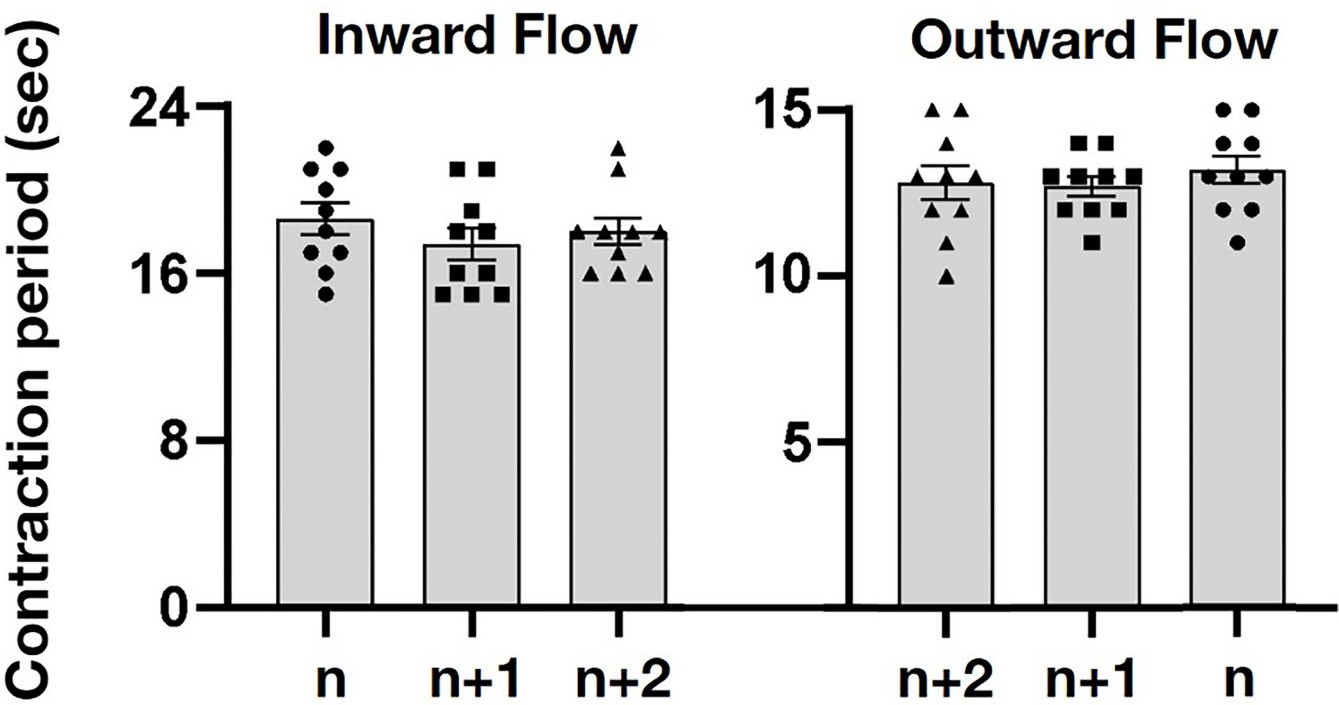

**Fig 7. Contraction period in consecutive order of the tract branches, n, n+1, and n+2 during the inward and outward flows.** The seconds of contraction in order 'n' are marked as circle dots, in 'n+1'are marked as solid squares, and in 'n+2' are marked as triangles. The ANOVA shows that designated inward flows have no significant differences in consecutive branches(n = 30, p = 0.50). The outward flows also show no significant differences between consecutive branches (n = 30, p = 0.67).

## Discussion

Over fifty species of polyclad flatworm can be found in the waters around Taiwan [27]. There has been a dearth of research on most of these relatively rare species; however, this study focused on *Paraplanocera oligoglena*, which is commonly found in intertidal zones throughout the region.

In our laboratory analysis of flatworms, we employed a bait that does not necessarily match the prey typically consumed by *P. oligoglena*; however, the widespread availability of clams in local markets made them a convenient substitute. The clams were also shown to extend the duration of digestion, which is crucial when seeking to observe the rate of predation on fresh bait and the bidirectional movements associated with digestion.

Wells and Sebens (2017) marked sea anemones by injecting them with methylene blue, neutral red, and fluorescein for laboratory analysis and field studies [19]. In the current study, we modified this procedure by using the same dyes to stain the clams consumed by *P. oligoglena*. Note that staining the food with fluorescein proved ineffective in enhancing the contrast between ROI and the rest of the body, even under blue light within a dark room. We also avoided neutral red during bait preparation due to its potential interference with the brownish patterns on the semi-transparent body. Thus, it was determined that methylene blue was the best option for staining the digestive tract. We anticipate that these dyeing methods could be used for future research on food chains in the wild.

In post-stain feed tracking, methylene blue dye proved highly effective in revealing both the widest (primary) tract as well as the narrowest (high-order) tract along this highly branched digestive system. ImageJ software proved largely ineffective in detecting the digestive pathway,

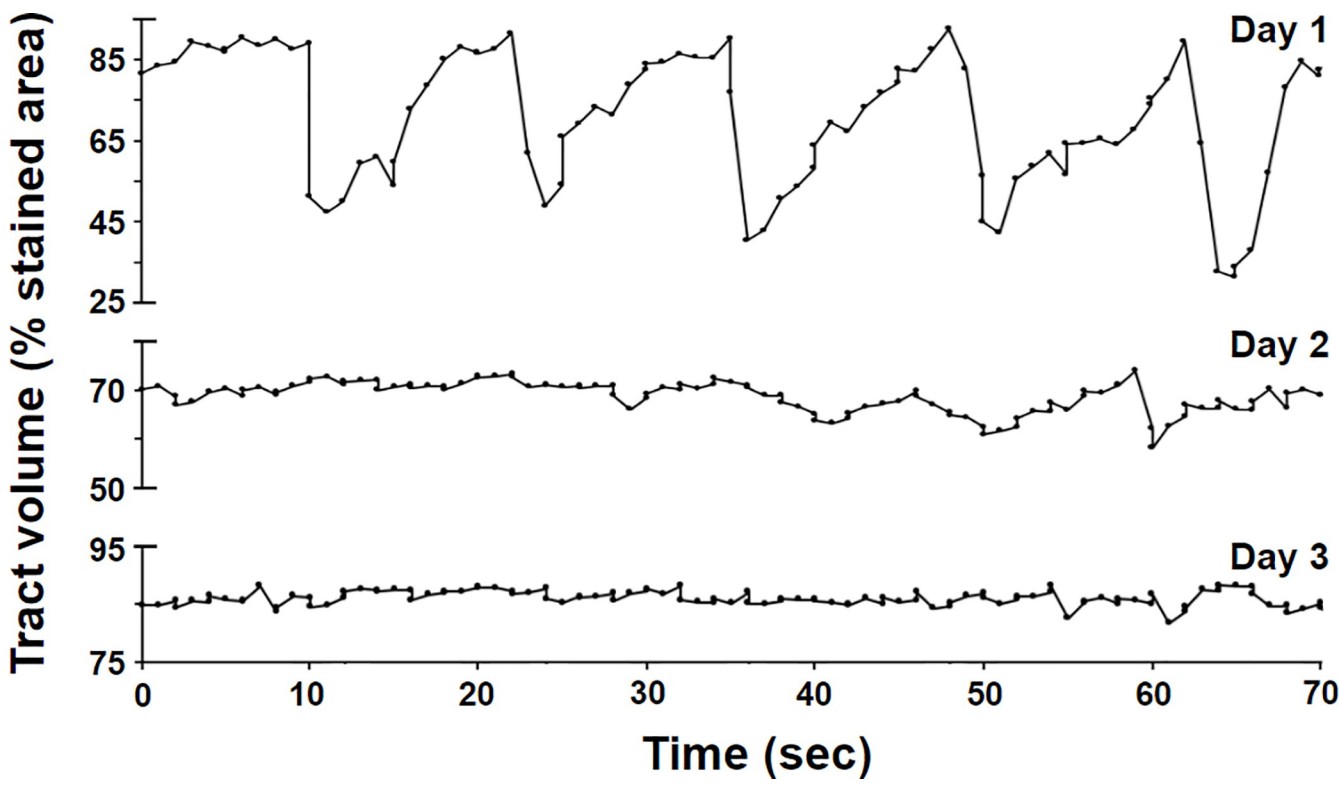

**Fig 8. Variations in tract volume on the 1st, 2nd, and 3rd days.**

due largely to the fact that many of the branches did not appear as continuous structures. Our use of adaptive thresholding in the image analysis of samples stained with methylene blue revealed continuous branches widely distributed across the entire body of the organisms. Note also that during the ingestion period, polyclad flatworms remain nearly stationary for extended durations of up to 6 hours [20,28]. This novel approach to analysis allowed tracking

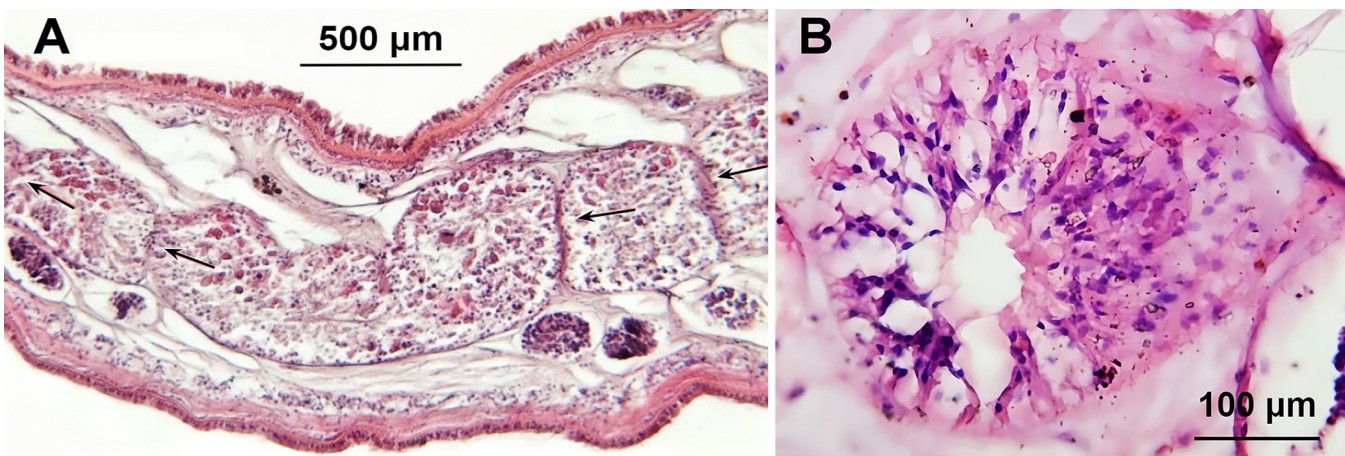

**Fig 9. Radial muscle within the digestive tract of *Paraplanocera oligoglena*.** (A) Longitudinal section highlighting the radial muscles with arrows. (B) Cross-section of a radial structure.

throughout this ingestion period in real-time or accelerated time. It is likely that this approach could be applied to other flatworm species in the future.

The sequentially bidirectional flow observed in *P. oligoglena* involves dual-directional peristalsis induced by radial muscles along the digestive tract. Newman and Cannon (2003) initially described the multibranched system of *Planocera* sp., suggesting the presence of small muscular valves along the branches [12]. In the current study, we identified radial muscle structures (Fig 9) distributed along the digestive branches. It appears that these structures regulate flow within the tract by providing waves of contractions to fill or clear the gut. This mechanism parallels previous descriptions of cnidarians [5]; however, it has not previously been identified in flatworms. Note that the transport of partially digested food back and forth along branches of various diameters at a constant speed suggests the corporations of the nerve system with those muscular valves along a tract during ingest process.

Koopowitz and Keenan (1982) claimed that flatworms are the most primitive animals with a true brain, based on the fact that an ipsilateral turn toward food requires intact connections between the brain and the main longitudinal nerve cords [29]. Accordingly, the feeding behavior of flatworms necessitates coordinated regulation between the brain and nervous system. If diffuse nerve net regulation is indeed required for feeding, then it can be inferred that contractions of the intestinal tract during ingestion may also be regulated by the nervous system. While the basis for the assumption about the role of the nervous system in regulating peristalsis in many branches of the digestive system is not entirely clear. Meanwhile the integral effect of the magistral hydroplasma flows in colonial hydroid is remaining one of the drive mechanism needed to be considered which might include dynamics of cell movement in the body, pulsations of hydrants and coenosarc, and movement of hydroplasma in the gastrovascular cavity [30].

A brain and nervous system with this degree of complexity should have the capacity to coordinate rhythmic contractions in the digestive tract more effectively than hydrozoans. Since the flatworms still exist on all levels of blind sacs distributed between its lowermost and highmost branches, the cause of peristaltic movement of flatworms may also bear the possibility of coordinating hydraulic relationships between different parts of the digestive tract. We infer that the peristaltic movement might function simultaneously with the nerve system and hydraulic relationships between different parts of the digestive tract, especially the hydroplasma flow in colonial hydroids have already been described by Marfenin and Dementyev (2024) in sufficient detail [30].

It is also possible that the movement of flatworms during ingestion could reveal the origin of sympathetic and parasympathetic nerve regulation. Contractions in the digestive tract during feeding are automatically regulated to facilitate the movement of food through the system. With the focus on the processing and movement of ingested food, it is not possible for the organism to exert voluntary control over these muscles during feeding. The regulation of muscular activity must wait until feeding is finished. These processes indicate the inhibition of sympathetic nerve regulation and the excitation of parasympathetic nerve regulation during ingestion.

The gastrovascular cavity functions as the primary organ of digestion and circulation in cnidarians and other animal phyla; however, those animals are unable to coordinate efficient digestion due to a lack of directional peristalsis. Previous laboratory studies on feeding revealed that these organisms enjoy a diverse diet, including various gastropods, such as sea snails and sea hares, and this broad range of food sources may contribute to the widespread distribution of flatworms in Taiwan. Bidirectional flow in the branched tracts of polyclads may provide survival advantages by allowing for digestion of greater efficiency. This might explain

why polyclad flatworms are larger and more mobile than other flatworms or lower organisms, such as cnidarians.

The cessation of digestion-related movement at day 2 after feeding provides evidence that the peristaltic and segmented movements primarily serve digestive functions, rather than circulatory functions (Fig 8). This may also explain why polyclad flatworms tend to be larger and more mobile than other flatworms [12] but still require an extremely thin body plan for efficient diffusion and interchange of materials with their environment.

The flatworms that received methylene blue (as indicated by the lingering presence of the dye) did not demonstrate any reluctance in subsequent feeding tests. This provides further evidence indicating the effectiveness of this marking scheme in identifying prey-predator relationships, particularly when implemented using the proposed image processing techniques aimed at enhancing contrast. In the future, these experiments should be performed in their natural habitat, such as tidal pools. These methods could also be used to elucidate the digestive mechanisms of other lower organisms.

## Supporting information

**S1 Video. Continuous video recording of *Paraplanocera oligoglena* after feeding with methylene blue dyed baits.**
(MP4)

**S2 Video. Time-lapse video converted from 120 seconds of continuous taping of the *Paraplanocera oligoglena* into 60 seconds clips after feeding with methylene blue dyed baits.**
(MP4)

**S1 Table. The data set of post-staining video tracking of dyed food passing through consecutive branches n, n+1, n+2, and n+3.** The acquired time of the events from the start (0.3 second) to the end (85.1 second).
(DOC)

**S2 Table. The data set of contraction periods in consecutive order of the tract branches, n, n+1, and n+2 during the inward and outward flows.**
(DOCX)

**S3 Table. The data set of variations in tract volume on the 1st, 2nd, and 3rd days.**
(DOCX)

## Acknowledgments

The authors express their sincere appreciation to Mr. Shang-Chi Wu for his generosity in collecting and transporting the flatworms. The authors are also deeply grateful to Mr. Shih-Chieh Kuo at the Chinese Culture University for specimen preservation.

## Author Contributions

**Conceptualization:** Wei-Ban Jie.

**Data curation:** Po-Chun Hsu.

**Formal analysis:** Yu-Ning Chiu.

**Investigation:** Po-Chun Hsu.

**Methodology:** Yu-Hsun Chang, Wei-Ban Jie.

**Software:** Yu-Hsun Chang.

**Supervision:** Wei-Ban Jie.

**Validation:** Po-Chun Hsu, Yu-Ning Chiu.

**Visualization:** Yu-Hsun Chang.

**Writing – original draft:** Po-Chun Hsu, Yu-Hsun Chang, Yu-Ning Chiu.

**Writing – review & editing:** Wei-Ban Jie.

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
