## [Decision Letter · Decision Letter 0]

20 Sep 2024

PONE-D-24-33814Sequentially bidirectional gastrovascular flows in intricately branched digestive tract of Planocerid flatwormsPLOS ONE

Dear Dr. Jie,

Thank you for submitting your manuscript to PLOS ONE. After careful consideration, we feel that it has merit but does not fully meet PLOS ONE’s publication criteria as it currently stands. Therefore, we invite you to submit a revised version of the manuscript that addresses the points raised during the review process.

We look forward to receiving your revised manuscript.

Kind regards,

Petr Heneberg

Academic Editor

PLOS ONE

Journal Requirements: When submitting your revision, we need you to address these additional requirements. 1. Please ensure that your manuscript meets PLOS ONE's style requirements, including those for file naming. The PLOS ONE style templates can be found at https://journals.plos.org/plosone/s/file?id=wjVg/PLOSOne_formatting_sample_main_body.pdf and https://journals.plos.org/plosone/s/file?id=ba62/PLOSOne_formatting_sample_title_authors_affiliations.pdf 2. PLOS requires an ORCID iD for the corresponding author in Editorial Manager on papers submitted after December 6th, 2016. Please ensure that you have an ORCID iD and that it is validated in Editorial Manager. To do this, go to ‘Update my Information’ (in the upper left-hand corner of the main menu), and click on the Fetch/Validate link next to the ORCID field. This will take you to the ORCID site and allow you to create a new iD or authenticate a pre-existing iD in Editorial Manager. 3. Please include your full ethics statement in the ‘Methods’ section of your manuscript file. In your statement, please include the full name of the IRB or ethics committee who approved or waived your study, as well as whether or not you obtained informed written or verbal consent. If consent was waived for your study, please include this information in your statement as well.

Reviewers' comments:

Reviewer's Responses to Questions

**Comments to the Author**

1. Is the manuscript technically sound, and do the data support the conclusions?

Reviewer #1: Yes

Reviewer #2: Partly

2. Has the statistical analysis been performed appropriately and rigorously? 

Reviewer #1: N/A

Reviewer #2: N/A

3. Have the authors made all data underlying the findings in their manuscript fully available?

Reviewer #1: Yes

Reviewer #2: No

4. Is the manuscript presented in an intelligible fashion and written in standard English?

Reviewer #1: Yes

Reviewer #2: Yes

5. Review Comments to the Author

Reviewer #1: A deceptively simple study that elucidates the in vivo functioning of the flatworm digestive tract. The authors develop a method that allows visualizing the gastrovascular system and examine the behavior of this system after feeding. The observed sequentially bidirectional flow is consistent with muscular action. (It might be worth mentioning in the introduction that simultaneously bidirectional flow suggests ciliary action, while sequentially bidirectional flow indicates muscular action.) Several comments and questions follow:

Line 92 - : What camera was used to capture images?

Line 192: Feeding apparently stimulates activity of the gastrovascular system. Hydroids behave similarly (see Dudgeon et al., 1999, Biol Bull 196, 1).

Line 292: Author contributions section is blank.

Reviewer #2: Review of manuscript #PONE-D-24-33814,

“Sequentially bidirectional gastrovascular flows in intricately branched digestive tract of Planocerid flatworms,” submitted by the authors for publication in PLOS ONE

The manuscript, in my opinion, must definitely be published, but first it needs to be substantially revised, correcting some omissions.

The manuscript submitted for publication in PLOS ONE describes the transport of food particles through the branched digestive system of one species of flatworms of the order Polycladida. Any data on the functioning of closed (not through) digestive systems are of great interest. The movement of food through a digestive tract that ends in a blind ends, rather than an anus, would theoretically seem impossible or at least difficult. Nevertheless, among invertebrates there are examples of successful solutions to this problem in nature. Some of these examples allow us to study non-centralized self-organization, when parts of the system are quite independent, and their interaction is equifinal in outcome.

Therefore, the research topic is promising, although the number of publications in this area is rare. It seems that the authors are not aware of some publications in which a similar phenomenon is described in colonial hydroids, whose digestive system is also quite extensive, and the movement of food particles is alternately bidirectional. A list of some publications is given below.

The authors have developed a suitable method for studying the functioning of the digestive system in a translucent, thin flatworm, and the results they obtained are impressive.

There are several questions regarding the contents of the manuscript.

The authors do not provide sufficient evidence to interpret the observed pulsations of the digestive tract segments as peristalsis. It is important to remember that a similar effect of transportation in other invertebrates with a pulsating digestive tract is achieved without peristalsis (see the recommended publications at the end of the review).

The description of food movement through the digestive tract is very superficial: without clarifications and quantitative indicators. The authors of the manuscript do not provide answers to the usual questions:

What is the speed of food movement from the pharynx to the distal parts of the digestive system?

How evenly and simultaneously does food move in different branches of the digestive system?

How equal in speed are the food supplies to the head and tail ends of the animal's body?

How do different branches of the digestive system relate to each other in terms of the speed of food movement?

The manuscript does not clearly indicate the samples. Or rather, there is no mention of samples at all.

The basis for the assumption about the leading role of the nervous system in regulating peristalsis in many branches (spurs) of the digestive system is not entirely clear. Thus, in the Discussion section (lines 252–255) it is written: “If diffuse nerve net regulation is indeed required for feeding, then it can be inferred that contractions of the intestinal tract during ingestion may also be regulated by the nervous system.

A brain and nervous system with this degree of complexity should have the capacity to coordinate rhythmic contractions in the digestive tract more effectively than hydrozoans».

And further (lines 255-256): "A brain and nervous system with this degree of complexity should have the capacity to coordinate rhythmic contractions in the digestive tract more effectively than hydrozoans". This statement does not take into account the possibility of coordinating local peristaltic activity without the participation of the nervous system, but only on the basis of hydraulic relationships between different parts of the digestive tract, especially since this has already been described in sufficient detail in hydroids. I believe that the authors should at least mention an alternative solution to the problem of transporting particles in closed systems.

I was unable to find a video in the material proposed for review, although the main conclusions of the future article are based on the video.

Further some comments.

Line or Fig. Question/Comment Explanation

Data availability The authors believe that “... all data are fully available without restriction ”, however, the manuscript lacks the primary data from which the authors arrived at the conclusions of the study.

There is no video, no choice with options for staining the subject after feeding, and no choice of digital data from which the graphs are plotted (Figures 6 and 7)

14 The reviewer did not find a video from which the authors drew important conclusions. How to check the objectivity of the authors if the material on which they base their conclusions is not presented?

96 - 118 There is no information on samples in the study of food particle transportation and in the study of histological preparations.

On the basis of what number of study objects (specimens of the species) are the results of the study derived? If only one specimen with the most pronounced process of food transport was used for the study, then this should be written about in the manuscript. In such a case, it is better for the researcher to use an Idiographic Approach (Marfenin, & Dementyev. 2022. Influence of Food Consumption...)

If conclusions are drawn from the study of several or many samples, the degree of variation in the process being studied should be clearly described in terms of the of peristalsis, regularity of contractions, amplitude of pulsation, etc..

116 What is the error of Matplotlib and Pandas methods ? Has the method been calibrated, i.e., verified its accuracy on images specially created by the authors for known volumes in advance ? Without specifying the error, any method, especially for pattern recognition, is questionable.

169 The authors should clarify the criteria on the basis of which they consider the pulsations they describe to be peristalsis.

How do the authors explain the difference in periods of retraction during pulsations of the digestive tract between inward flows and outward flows ? ( see Fig . 7)

361 Why does the caption to the picture #1 say specimens and not specimen ? How many samples did you study in total? The manuscript nowhere states the number of samples used in the study.

The manuscript does not end with clear conclusions, i.e. proven statements. For this article, conclusions section would be very relevant.

375 The figure caption does not explain the numerical designations. It is not clear whether these designations refer to a segment of the digestive tract with several sections or only to an individual section of the segment.

Fig. 5 Figure 5 is not sufficient to prove or even illustrate peristalsis. The same effect of pulsations of the digestive tract can occur without peristalsis, as, for example, in colonial hydroids.

Fig. 6 Is it correct to call the “Y” axis “Tract volume” when in fact it is “% of the colored part of the digestive trust” ?

Fig. 7 The caption to Figure 7 clearly lacks information: there is no explanation of the symbols: circles, squares, triangles, Could it be variation in sampling? Then it seems to be the only indication of sampling in the manuscript.

Suggested publications on the topic of the manuscript for comparison of two possible mechanisms of food transport in a closed digestive system:

Marfenin, N.N., Dementyev, V.S. Integral Effect of Interaction of Parts of a Noncentralized Biosystem by the Example of Magistral Hydroplasma Flow Formation in the Shoots of Colonial Hydroid Dynamena pumila (L., 1758). Biol Bull Rev 14, 344–359 (2024). https://doi.org/10.1134/S207908642403006X

Dementyev, V.S., Marfenin, N.N. Express Transport of Particles in the Stolon of the Colonial Hydroid Dynamena pumila (L., 1758). Biol Bull Rev 13, 9–19 (2023). https://doi.org/10.1134/S2079086423010024

Marfenin, N.N., Dementyev, V.S. Influence of Food Consumption on the Functioning of the Pulsator-Reversible Transport System in Hydroids—An Idiographic Approach. Biol Bull Rev 12, 483–503 (2022). https://doi.org/10.1134/S207908642205005X

Dementyev, V.S., Marfenin, N.N. Efficiency of the Transport System of the Hydroid Dynamena pumila (L., 1758) under Different Abiotic Impacts. Biol Bull Rev 12, 266–278 (2022). https://doi.org/10.1134/S2079086422030021

Marfenin, N.N., Dementyev, V.S. Paradox of Extended Flows in Dynamena pumila (Linnaeus, 1758) Colonial Hydroid. Biol Bull Rev 8, 212–226 (2018). https://doi.org/10.1134/S2079086418030088

Marfenin, N.N., 2016. Decentralized Organism Exemplified with Colonial Hydroid Species. Biosphere 8, 315–337 (2016). http://dx.doi.org/10.24855/biosfera.v8i3.264 second link: https://www.researchgate.net/publication/331716150_DECENTRALIZED_ORGANISM_EXEMPLIFIED_WITH_COLONIAL_HYDROID_SPECIES_NN_Marfenin

I believe that after the manuscript has been revised, it can be recommended for publication.

6. PLOS authors have the option to publish the peer review history of their article (what does this mean?). If published, this will include your full peer review and any attached files.

Reviewer #1: No

Reviewer #2: No

---

## [Author Response · Author response to Decision Letter 0]

13 Nov 2024

We are very appreciative of the reminder offered by reviewers. We have read through those references and carefully discussed them. A lot of revisions are based on whose perspective aside just from polyclads'. Thanks.

---

## [Decision Letter · Decision Letter 1]

3 Dec 2024

Sequentially bidirectional gastrovascular flows in intricately branched digestive tract of planocerid flatworms

PONE-D-24-33814R1

Dear Dr. Jie,

We’re pleased to inform you that your manuscript has been judged scientifically suitable for publication and will be formally accepted for publication once it meets all outstanding technical requirements.

Kind regards,

Petr Heneberg

Academic Editor

PLOS ONE

Additional Editor Comments (optional):

Reviewers' comments:

Reviewer's Responses to Questions

**Comments to the Author**

1. If the authors have adequately addressed your comments raised in a previous round of review and you feel that this manuscript is now acceptable for publication, you may indicate that here to bypass the “Comments to the Author” section, enter your conflict of interest statement in the “Confidential to Editor” section, and submit your "Accept" recommendation.

Reviewer #1: All comments have been addressed

Reviewer #2: (No Response)

2. Is the manuscript technically sound, and do the data support the conclusions?

Reviewer #1: Yes

Reviewer #2: Yes

3. Has the statistical analysis been performed appropriately and rigorously? 

Reviewer #1: N/A

Reviewer #2: Yes

4. Have the authors made all data underlying the findings in their manuscript fully available?

Reviewer #1: Yes

Reviewer #2: Yes

5. Is the manuscript presented in an intelligible fashion and written in standard English?

Reviewer #1: Yes

Reviewer #2: Yes

6. Review Comments to the Author

Reviewer #1: As I pointed out in my previous review this is a deceptively simple study that elucidates the in vivo functioning of the flatworm digestive tract. The authors develop a method that allows visualizing the gastrovascular system and examine the behavior of this system after feeding. The observed sequentially bidirectional flow is consistent with muscular action, and muscular valves are identified. I find the revisions suitable and have no further concerns except the following very small ones:

Line 60: “Planocerid”: This is not a formal taxonomic name. So, it should not be capitalized or italicized.

Lines 278-281: “Since the flatworms still exist on all levels of blind sacs distributed between its lowermost and highmost branches, the cause of peristaltic movement of flatworms may also bear the possibility of coordinating hydraulic relationships between different parts of the digestive tract.”: The writing of this sentence is confusing. Minimally, it seems that it should be “highermost.”

Reviewer #2: The authors have taken into account a significant part of the comments. At present, the manuscript more closely meets the requirements for the publication of scientific articles and may be preferable for printing.

It is a pity that the authors did not provide a table with responses to the reviewer's comments. This is not typical when correcting a manuscript and partially complicates the reviewer's work and increases the likelihood of misunderstanding each other.

The absence of conclusions to the article obviously reduces its significance. Conclusions do not repeat the abstract or summary, but present the results in a more rigorous form.

7. PLOS authors have the option to publish the peer review history of their article (what does this mean?). If published, this will include your full peer review and any attached files.

Reviewer #1: No

Reviewer #2: No

---

## [Editor Report · Acceptance letter]

9 Dec 2024

PONE-D-24-33814R1 

PLOS ONE

Dear Dr. Jie, 

I'm pleased to inform you that your manuscript has been deemed suitable for publication in PLOS ONE. Congratulations! Your manuscript is now being handed over to our production team.

Kind regards, 

on behalf of

Dr. Petr Heneberg 

Academic Editor

PLOS ONE